# Design, Characterization, and Anticancer and Antimicrobial Activities of Mucoadhesive Oral Patches Loaded with *Usnea barbata* (L.) F. H. Wigg Ethanol Extract F-UBE-HPMC

**DOI:** 10.3390/antiox11091801

**Published:** 2022-09-13

**Authors:** Violeta Popovici, Elena Matei, Georgeta Camelia Cozaru, Laura Bucur, Cerasela Elena Gîrd, Verginica Schröder, Emma Adriana Ozon, Mirela Adriana Mitu, Adina Magdalena Musuc, Simona Petrescu, Irina Atkinson, Adriana Rusu, Raul-Augustin Mitran, Mihai Anastasescu, Aureliana Caraiane, Dumitru Lupuliasa, Mariana Aschie, Eugen Dumitru, Victoria Badea

**Affiliations:** 1Department of Microbiology and Immunology, Faculty of Dental Medicine, Ovidius University of Constanta, 7 Ilarie Voronca Street, 900684 Constanta, Romania; 2Center for Research and Development of the Morphological and Genetic Studies of Malignant Pathology, Ovidius University of Constanta, CEDMOG, 145 Tomis Blvd., 900591 Constanta, Romania; 3Clinical Service of Pathology, Sf. Apostol Andrei Emergency County Hospital, 145 Tomis Blvd., 900591 Constanta, Romania; 4Department of Pharmacognosy, Faculty of Pharmacy, Ovidius University of Constanta, 6 Capitan Al. Serbanescu Street, 900001 Constanta, Romania; 5Department of Pharmacognosy, Phytochemistry, and Phytotherapy, Faculty of Pharmacy, Carol Davila University of Medicine and Pharmacy, 6 Traian Vuia Street, 020956 Bucharest, Romania; 6Department of Cellular and Molecular Biology, Faculty of Pharmacy, Ovidius University of Constanta, 6 Capitan Al. Serbanescu Street, 900001 Constanta, Romania; 7Department of Pharmaceutical Technology and Biopharmacy, Faculty of Pharmacy, Carol Davila University of Medicine and Pharmacy, 6 Traian Vuia Street, 020956 Bucharest, Romania; 8Ilie Murgulescu Institute of Physical Chemistry, Romanian Academy, 202 Spl. Independentei, 060021 Bucharest, Romania; 9Department of Oral Rehabilitation, Faculty of Dental Medicine, Ovidius University of Constanta, 7 Ilarie Voronca Street, 900684 Constanta, Romania; 10Department of Gastroenterology, Emergency Hospital of Constanța, 145 Tomis Blvd., 900591 Constanta, Romania

**Keywords:** *Usnea barbata* dry ethanol extract, oral mucoadhesive patches, physico-chemical properties, pharmacotechnical properties, oral health, oral cancer, anticancer activity, ROS, antimicrobial activity

## Abstract

The oral cavity’s common pathologies are tooth decay, periodontal disease, and oral cancer; oral squamous cell carcinoma (OSCC) is the most frequent oral malignancy, with a high mortality rate. Our study aims to formulate, develop, characterize, and pharmacologically investigate the oral mucoadhesive patches (F-UBE-HPMC) loaded with *Usnea barbata* (L.) F.H. Wigg dry ethanol extract (UBE), using HPMC K100 as a film-forming polymer. Each patch contains 312 µg UBE, with a total phenolic content (TPC) of 178.849 µg and 33.924 µg usnic acid. Scanning electron microscopy (SEM) and atomic force microscopy (AFM) were performed for their morphological characterization, followed by Fourier transform infrared spectroscopy (FTIR), X-ray diffraction (XRD), and thermogravimetric analysis (TGA). Pharmacotechnical evaluation involved the measurement of the specific parameters for mucoadhesive oral patches as follows: weight uniformity, thickness, folding endurance, tensile strength, elongation, moisture content, pH, disintegration time, swelling rate, and ex vivo mucoadhesion time. Thus, each F-UBE-HPMC has 104 ± 4.31 mg, a pH = 7.05 ± 0.04, a disintegration time of 130 ± 4.14 s, a swelling ratio of 272 ± 6.31% after 6 h, and a mucoadhesion time of 102 ± 3.22 min. Then, F-UBE-HPMCs pharmacological effects were investigated using brine shrimp lethality assay (BSL assay) as a cytotoxicity prescreening test, followed by complex flow cytometry analyses on blood cell cultures and oral epithelial squamous cell carcinoma CLS-354 cell line. The results revealed significant anticancer effects by considerably increasing oxidative stress and blocking DNA synthesis in CLS-354 cancer cells. The antimicrobial potential against *Staphylococcus aureus* ATCC 25923, *Pseudomonas aeruginosa* ATCC 27353, *Candida albicans* ATCC 10231, and *Candida parapsilosis* ATCC 22019 was assessed by a Resazurin-based 96-well plate microdilution method. The patches moderately inhibited both bacteria strains growing and displayed a significant antifungal effect, higher on *C. albicans* than on *C. parapsilosis*. All these properties lead to considering F-UBE-HPMC suitable for oral disease prevention and therapy.

## 1. Introduction

Oral health consists of the health of the oral mucosa, gums, teeth, and the whole system that allows us to speak, chew, taste, swallow, smile, and communicate with confidence various emotions through facial expressions. It is an essential component of general health, implying physiological, social, and psychological attributes fundamental to the quality of life [1]. The oral cavity’s common pathologies group includes dental decays, gums and periodontia diseases, and oral cancer; oral squamous cell carcinoma (OSCC) is a substantially frequent oral malignancy (over 80–90% of all oral cavity’s neoplasms) [2].

Oral diseases are caused by different infectious pathogens [3,4] and modifiable risk factors, including poor hygiene, sugar consumption, tobacco use, alcohol use, and other damaging habits [5]; various social and commercial determinants influence their impact. According to a World Health Organization (WHO) resolution [6], numerous oral health conditions could be preventable or treated in the early stages. In addition to conventional therapy, many phytotherapeutic products are made, especially for prophylaxis and early treatment of oral cavity diseases [7]. These products have various pharmaceutical formulations, as follows: concentrated mouthwashes, sprays, badinages, and mucoadhesive patches with antimicrobial and anti-inflammatory activities. Recently, Kumar et al. [8] detailed the benefits of apitherapy in periodontal diseases based on in vitro, in vivo, and clinical studies. They also profoundly analyzed the significant role of plant secondary metabolites with antioxidant potential in maintaining oral health [9]. Prakash et al. [10] extensively reviewed plant-based antioxidant extracts and compounds to manage oral cancer.

In the plant world, lichens represent a promising source of anticancer and antibiotic drugs [11]. Numerous studies have analyzed the cytotoxic and anticancer properties of *Usnea* sp. extracts [12,13,14,15,16,17,18,19,20], isolated metabolites, and their derivatives [21,22,23,24,25,26,27,28]. Some authors explored the potential pharmaceutical applications of usnic acid as an anticancer drug, aiming to increase its biodisponibility and simultaneously diminish its most known toxicity [29]. The usnic acid medicinal application limits, such as low solubility in water, hepatotoxicity, and reduced therapeutical index, have led to innovative pharmaceutical formulations [30] with antitumor effects. Thus, Alpsoy et al. [31] developed stable usnic acid (UA)-conjugated superparamagnetic iron oxide nanoparticles (SPIONs) as a potential drug carrier for in vitro analysis of MCF-7 (breast cancer), HeLa (cervix cancer), L-929 (mouse fibroblast), U-87 (glioblastoma), and A-549 (human lung cancer) cell lines. Pereira da Silva Santos et al. [32] investigated the antitumor activity of usnic acid encapsulated into nanocapsules prepared with lactic-co-glycolic acid polymer. Garg et al. [33] formulated usnic acid (UA) loaded heparin-modified gellan gum (HAG) nanoparticles (NPs), proving their substantial antitumor potential on A-549 cancer cells. However, data regarding the anticancer activity of different pharmaceutical formulations with *Usnea* sp extracts found in the accessed scientific literature are very rare; Isik et al. reported the anticancer activity of *Usnea* sp. extract-based synthesized Ag@ZnO bimetallic nanocomposite on the H-SY5Y human neuroblastoma cell line [34].

Most lichen-based nanoparticles (NPs) have antibacterial effects [35]. Few researchers designed different pharmaceutical formulations with *Usnea* lichens and evaluated their antibacterial effects. Thus, Siddiqi et al. [36] described the biogenic fabrication and characterization of silver nanoparticles using *U. longissima* aqueous-ethanolic extract and analyzed their antibacterial activity. They reported the nanoparticles’ inhibitory effect on *S. aureus*, *P. aeruginosa,* and other Gram-positive and Gram-negative bacteria due to silver ions released from Ag NPs. Moreover, they suggested the following four possible bactericidal mechanisms: interference during cell wall synthesis, protein biosynthesis suppression, disruption of the transcription processes, and major metabolic pathways. Abdolmaleki et al. [37] obtained silver nanoparticles from two lichens (*U. articulata* and *R. sinensis*) with effective antibacterial activity against *S. aureus* and *P. aeruginosa.* Balaz et al. [38] recently proposed a bio-mechanochemical synthesis of silver nanoparticles using *U. antarctica* and other lichen species. Using AgNO_3_ (as a silver precursor) and lichens (as reduction agents), they performed techniques of mechanochemistry (ball milling) and obtained nanoparticles with an intense antibacterial effect against *S. aureus*. This described procedure overcomes the lichen secondary metabolites’ low solubility in water [38].

Tadic et al. [39] performed an oral product formulated as compressed tablets based on plant extracts/essential oils. It contains *U. barbata* supercritical CO_2_ extract, *O. heracleoticum* L. and *Sideritis scardica* L. water-alcohol extracts, and *Satureja montana* L. essential oil. The stated oral formulation is intended for topical application (local treatment of the inflammation of oropharyngeal mucosa), comprising a combination of herbal preparations with antimicrobial activity against causative agents. We aim to formulate and develop mucoadhesive oral patches loaded with *Usnea barbata* (L.) F. H. Wigg dry ethanol extract and evaluate their cytotoxicity and in vitro anticancer and antimicrobial activities. Our results suggest that this lichen-based pharmaceutical formulation is suitable for potential use in various oral disease therapies.

## 2. Materials and Methods

### 2.1. Materials

Our study’s chemicals, standards, and reagents were of analytical grade. Usnic acid standard 98.1% purity, Propidium Iodide (PI) 1.0 mg/mL, Dimethyl sulfoxide (DMSO), Polyethylene Glycol 400 (PEG 400), and Hydroxypropyl methylcellulose (HPMC) and Antibiotics mix solution—100 µL/mL with 10 mg Streptomycin, 10,000 U Penicillin, 25 µg Amphotericin B per 1 mL—were provided by Sigma-Aldrich Chemie GmbH (Taufkirchen, Germany). Annexin V Apoptosis Detection Kit and flow cytometry staining buffer (FCB) were purchased from eBioscience^TM^ (Frankfurt am Main, Germany) and RNase A 4 mg/mL from Promega (Madison, WI, USA). Magic Red^®^ Caspase-3/7 Assay Kit, Reactive Oxygen Species (ROS) Detection Assay Kit, and EdU i-Fluor 488 Kit were supplied by Abcam (Cambridge, UK).

The OSCC cell line (CLS-354) and the culture medium—Dulbecco’s Modified Eagle’s Medium (DMEM) High Glucose, basic supplemented with 4.5 g/L glucose, L-glutamine and 10% Fetal Bovine Serum (FBS) were provided by CLS Cell Lines Service GmbH (Eppelheim, Germany). Trypsin-ethylenediamine tetra acetic acid (Trypsin EDTA) and the media for blood cells—Dulbecco’s phosphate-buffered saline with MgCl_2_ and CaCl_2_, FBS and L-Glutamine (200 mM) solution—were purchased from Gibco^TM^ Inc (Billings, MT, USA).

From a non-smoker healthy donor (BIII, Rh+), the blood samples were collected according to Ethical approval code 7080/10.06.2021 from Ovidius University of Constanta and Donor Consent code 39/30.06.2021.

*U. barbata* was harvested in March 2021 from the forest in the Călimani Mountains (47°29′ N, 25°12′ E, and 900 m altitude) and identified by the Department of Pharmaceutical Botany of the Faculty of Pharmacy, Ovidius University of Constanta, using standard methods. A voucher specimen is deposited in the Herbarium of Pharmacognosy Department, Faculty of Pharmacy, Ovidius University of Constanta (Popovici 3/2021, Ph-UOC). The 96% ethanol for *U. barbata* dry extract preparation was provided by Chimreactiv SRL Bucharest, Romania.

*Artemia salina* eggs and *Artemia* salt (Dohse Aquaristik GmbH & Co. Gelsdorf, Germany) were purchased online from https://www.aquaristikshop.com/ (accessed on 5 May 2022).

Bacterial and fungal cell lines (*S. aureus* ATCC 25923, *P. aeruginosa* ATCC 27353, *C. albicans* ATCC 10231, and *C. parapsilosis* ATCC 22019) for antimicrobial activity evaluation were obtained from Microbiology Department, S.C. Synevo Romania S.R.L., Constanta Laboratory according to agreement of partnership No 1060/25.01.2018 with the Faculty of Pharmacy, Ovidius University of Constanta. Thermo Fisher Scientific (GmbH, Dreieich, Germany) provided culture medium Mueller-Hinton agar (MHA); Resazurin solution (from in vitro Toxicology Assay Kit, TOX8-1KT, Resazurin based), and RPMI 1640 Medium were purchased from Sigma-Aldrich Chemie GmbH (Taufkirchen, Germany).

### 2.2. Formulation and Development of Mucoadhesive Oral Patches

The *U. barbata* dry extract in ethanol (UBE) was obtained through a method described in a previous study [40]. The dried lichen was ground in a laboratory mill, LM 120 (Perkin Elmer, Waltham, MA, USA) [41], and extracted for eight hours with 96% ethanol in a Soxhlet continuous reflux system. The Soxhlet extraction was performed at the ethanol boiling point (65–70 °C). The rotary evaporator TURBOVAP 500 Caliper was used for solvent evaporation. Next, the extract was kept for 16 h in a chemical exhaust hood for optimal solvent evaporation. The obtained dry extract was transferred to a sealed glass bottle and stored in the freezer (Sirge FREEZER) at −24 °C until processing [40].

To develop the mucoadhesive oral patches containing *U. barbata* dry extract in 96% ethanol (F-UBE-HPMC), we used HPMC K100 with a viscosity of 100 mPa as a film-forming polymer and PEG 400 as an external plasticizer for its high hydrophilic character and non-toxicity [42].

To prove the UBE activity and influence on the F-UBE-HPMC patches’ pharmaceutical characteristics, we prepared mucoadhesive oral patches containing suitable excipients but no active ingredient load. We used them as References (R).

The UBE amount was selected to achieve a suitable dosage between the formulations. HPMC was weighed using a Mettler Toledo AT261 balance (Marshall Scientific, Hampton, NH, USA) with 0.01 mg sensitivity for the polymeric matrix system. Then, it was dispersed in water by stirring at 700 rpm and room temperature, using an MR 3001K magnetic stirrer (Heidolph Instruments GmbH & Co. KG, Schwabach, Germany); PEG 400 was added and mixed. UBE was dissolved in ethanol, and the solution was slowly added to the previously prepared matrix and stirred in the same conditions. Reference products were realized by mixing the 96% ethanol with the base system.

The formed gels were left overnight at room temperature for deaeration. The viscous dispersions were poured in a thin layer into Petri glass plates and dried in ambient conditions for 24 h. Finally, the dried patches were peeled off the plate surface and cut into (1.5 × 2) cm patches.

### 2.3. Physico-Chemical Analysis of Mucoadhesive Oral Patches

#### 2.3.1. SEM Morphology

The mucoadhesive patches morphology was investigated by scanning electron microscopy (SEM) in a high-resolution scanning electron microscope Quanta3D FEG (Thermo Fisher Scientific, GmbH, Dreieich, Germany).

#### 2.3.2. Atomic Force Microscopy 

The atomic force microscopy (AFM) measurements were carried out with AFM XE-100 (Park Systems Corporate, Suwon, Korea). AFM images were obtained in non-contact mode to minimize the tip-sample interaction; the microscope was equipped with flexure-guided, crosstalk-eliminated scanners. They were registered with sharp tips (PPP-NCLR, from NANOSENSORS™, Neuchatel, Switzerland) with less than 10-nm radius of curvature, 225 mm mean length, 38 mm mean width, ~48 N/m force constant, and a resonance frequency of 190 kHz. The AFM image processing was performed with an XEI program (v 1.8.0—Park Systems Corporate, Suwon, Korea) to display purpose and to evaluate roughness. Representative line scans are presented below the images in so-called “enhanced contrast” mode, showing the surface profile of the scanned samples (the dimensions of the selected particles are indicated with red arrows along the fixed line).

#### 2.3.3. Fourier Transform Infrared Spectroscopy

Fourier transform infrared (FTIR) spectra of both patches were recorded using a Nicolet Spectrometer 6700 FTIR (Thermo Electron Corporation, Waltham, MA, USA) apparatus with a Smart DuraSamplIR HATR (Horizontal Attenuated Total Reflectance) accessory and a laminated–diamond crystal in the range of 400–4000 cm^−1^, in transmittance mode.

#### 2.3.4. X-ray Diffraction

X-ray Diffraction (XRD) patterns were registered using a Rigaku Ultima IV diffractometer (Rigaku Corporation, Tokyo, Japan) in parallel beam geometry with a step size of 0.02 and a speed of 2° (2θ)/min over a range of 5–60°. A CuKα tube (λ = 1.54056 Å) operating at 40 kV and 30 mA was the source of the X-rays.

#### 2.3.5. Thermogravimetric Analysis

Thermogravimetric analysis (TGA) coupled with differential thermal analyses (DTA) was performed using a Mettler Toledo TGA/SDTA851e thermogravimetric analyzer (Mettler-Toledo GmbH, Greifensee, Switzerland), at a heating rate of 10 °C min^−1^, under 80 mL min^−1^ synthetic air atmosphere.

### 2.4. Pharmacotechnical Analysis of Mucoadhesive Oral Patches

#### 2.4.1. Weight Uniformity

The weight uniformity was evaluated on 20 patches of both formulations (F-UBE-HPMC and Reference). They were individually weighed, and the average weight was determined.

#### 2.4.2. Thickness

This parameter was also measured on 20 patches of each formulation (F-UBE-HPMC and Reference) using a Yato digital micrometer (Yato China Trading Co., Ltd., Shanghai, China) with a 0–25 mm measuring range and 0.001 mm resolution. The mean value was calculated.

#### 2.4.3. Folding Endurance

The F-UBE-HPMC patches were repeatedly folded and rolled until they broke, or up to 300× [43]. The folding times were registered and expressed as folding endurance values.

#### 2.4.4. Tensile Strength and Elongation Ability

The tensile strength and elongation ability were determined using an LR 10K Plus digital tensile force tester for universal materials (Lloyd Instruments Ltd., West Sussex, UK). The analysis was performed from a 30 mm distance with a speed of 30 mm/min. The patch was placed vertically between the two braces, and the breakage force was registered. The measurement was executed in triplicate.

The following equations (Equations (1) and (2)) were used to calculate the tensile strength and the elongation at break:(1)$$\mathrm{Tensile}\;\mathrm{strength}\;\left(\mathrm{kg}/{\mathrm{mm}}^{2}\right)=\frac{\mathrm{Force}\;\mathrm{at}\;\mathrm{breakage}\;\left(\mathrm{kg}\right)}{\mathrm{Patch}\;\mathrm{thickness}\;\left(\mathrm{mm}\right)\times \;\mathrm{Patch}\;\mathrm{width}\;\left(\mathrm{mm}\right)}$$
(2)$$\mathrm{Elongation}\;\%\;=\frac{\mathrm{Increase}\;\mathrm{in}\;\mathrm{patch}\;\mathrm{length}\;\left(\mathrm{cm}\right)}{\mathrm{Initial}\;\mathrm{patch}\;\mathrm{length}\;\left(\mathrm{cm}\right)}\times 100$$

#### 2.4.5. Moisture Content

The moisture content was assessed as the loss on drying by the thermogravimetric method using an HR 73 halogen humidity analyzer (Mettler-Toledo GmbH, Greifensee, Switzerland) [6]. Five patches of each formulation were analyzed.

#### 2.4.6. Surface pH

Five patches of each formulation were moistened with 1 mL of distilled water (pH 6.5 ± 0.5) for 5 min at room temperature. The pH value was determined by touching the electrode of the CONSORT P601 pH-meter (Consort bvba, Turnhout, Belgium) with the patch surface.

#### 2.4.7. In Vitro Disintegration Time

The time required to disintegrate the patches, with no residual mass thoroughly, was measured in simulated saliva phosphate buffer pH of 6.8 at 37 ± 2 °C, using an Erweka DT 3 apparatus (Erweka^®^ GmbH, Langen, Germany) [44].

#### 2.4.8. Swelling Ratio

Six patches of each formulation were placed on 1.5% agar gel in Petri plates and incubated at 37 ± 1 °C. Every 30 min, for 6 h, the patches were weighed. The swelling ratio was calculated according to Equation (3) as follows:(3)$$\mathrm{Swelling}\;\mathrm{ratio}\;=\frac{\mathrm{Wt}-\mathrm{Wi}}{\mathrm{Wi}}\times 100$$
Wt—the patch’s weight at “t” time after incubation and Wi—the initial weight [43,45,46,47].

#### 2.4.9. Ex Vivo Mucoadhesion Time

The study was performed by the method described by Gupta et al. [48] on a detached porcine buccal mucosa; then, the fat layer and any tissue residue were removed. The membrane was washed with ultrapure water and a phosphate buffer pH 6.8 at 37 °C, then fixed on a glass plate. Each F-UBE-HPMC patch was hydrated in the center with 15 μL phosphate buffer and brought to the mucosa surface by pressing it for 30 s. The glass plate was placed in 200 mL phosphate buffer pH 6.8 and maintained at 37 °C for 2 min. A paddle with a stirring rate of 28 rpm was operated to ensure the appropriate simulation of the oral cavity conditions.

The mucoadhesion time was expressed as the time needed by each patch to detach over the oral mucosa. All tests were realized in triplicate.

### 2.5. Evaluation of the Cytotoxic Activity of Mucoadhesive Oral Patches on A. salina Larvae

#### 2.5.1. Sample Preparation

F-UBE-HPMC was placed in a diluted buffer (1 mL) and incubated for 15 min at 37 °C; then, its homogenous dispersion in the buffer solution was observed.

#### 2.5.2. BSL Assay

*Artemia salina* (brine shrimp) was used as an animal model for the F-UBE-HPMC cytotoxicity investigation, adapted from Nazir et al. [49]. The *A. salina* larvae were obtained under continuous light and aeration conditions at a temperature of 20 °C by introducing the cysts for 24–48 h in a saline solution of 0.35%. The brine shrimp larvae in the first stage (instar I) were introduced in 0.3% saline solution into experimental pots (with a volume of 1 mL) [50]. The analysis was compared to a blank (untreated nauplii) to obtain accurate results regarding the F-UBE-HPMC cytotoxic effect. The nauplii were not fed during the test to not interfere with the tested extracts. Their evolution was evaluated after 24 h and 48 h; the larvae had embryonic energy reserves as lipids throughout this period.

#### 2.5.3. Fluorescent Microscopy

The brine shrimp larvae were stained with 3% acridine orange (Merck Millipore, Burlington, MA, USA) for 5 min. The samples were subjected to drying for 15 min in darkness and placed on the microscope slides.

#### 2.5.4. Data Processing

The microscopic images were achieved using a VWR microscope VisiScope 300D (VWR International, Radnor, PA, USA) with a Visicam X3 camera (VWR International Radnor, PA, USA) at 40×, 100×, and 400× magnifications and processed with VisiCam Image Analyzer 2.13.

Fluorescent microscopy images were obtained using an OPTIKA B-350 microscope (Ponteranica, BG, Italy) blue filter (λex = 450–490 nm; λem = 515–520 nm) and green filter (λex = 510–550 nm; λem = 590 nm) [41]. The FM images at 100× and 400× magnification were processed with Optikam Pro 3 Software (OPTIKA S.R.L., Ponteranica, BG, Italy). 

All observations were performed in triplicate.

### 2.6. In Vitro Analysis of the Biological Effects of Mucoadhesive Oral Patches on Human Blood Cell Cultures and Oral Cancer Cell Line CLS-354 

#### 2.6.1. Equipment

Our study platform for in vitro cytotoxicity analysis of F-UBE-HPMC was the Attune Acoustic focusing cytometer (Applied Biosystems, Bedford, MA, USA). Before cell analysis, the flow cytometer was first set by using fluorescent beads—Attune performance tracking beads, labeling, and detection (Life Technologies, Europe BV, Bleiswijk, The Netherlands) [51], with a standard size (four intensity levels of beads population). The cell number was established by enumerating cells below 1 µm [52]. Using Forward Scatter (FSC) and Side Scatter (SSC), more than 10,000 cells per sample for each analysis were gated.

#### 2.6.2. Data Processing

Flow cytometry data were achieved using Attune Cytometric Software v.1.2.5, Applied Biosystems 2010 (Bedford, MA, USA).

#### 2.6.3. Human Blood Cells Cultures

The blood sample was collected into heparin vacutainers. The heparinized blood (1.0 mL) was added to untreated Nunclon Vita Cell culture 6-well plates (Kisker Biotech GmbH & Co.KG, Steinfurt, Germany), together with 6.0 mL of Dulbecco’s phosphate-buffered saline with MgCl_2_ and CaCl_2_ medium supplemented with 10% bovine fetal serum, L-glutamine, and antibiotic mix solution. They were incubated in a Steri-Cycle™ i160 CO_2_ Incubator (Thermo Fisher Scientific Inc, Waltham, MA, USA) with 5% CO_2_ at 37 °C. After 72 h, the blood cell cultures were treated with the samples and controls. Then, the cells were subjected to 24 h of incubation under the same conditions [51]. All flow cytometry analyses were performed after this incubation time.

#### 2.6.4. CLS-354 Cell Line, Cells Culture

The CLS-354 tumor cells were cultured in DMEM High Glucose with 10% FBS supplemented with antibiotic mix solution in humidity conditions of 5% CO_2_ at 37 °C for 7 days [53]. Then, the cells were dissociated with Trypsin-EDTA and centrifugated at 3000 rpm for 10 min in a Fisher Scientific GT1 Centrifuge (Thermo Fisher Scintific Inc, Waltham, MA, USA). Then, the cells were distributed in Millicell™ 24-Well Cell Culture Microplates (Termo Fisher Scientific Inc, Waltham, MA, USA). After treatment, the cells were incubated for 24 hours in the same conditions [54]. All the flow-cytometry analyses were performed after this incubation period.

#### 2.6.5. Samples and Control Solutions

The F-UBE-HPMC were dissolved in the suitable culture media for both types of cells with 1% DMSO. As a positive control, usnic acid (125 µg/mL in 1% DMSO) was selected, and the negative control was 1% DMSO.

### 2.7. Evaluation of Total ROS Activity

In total, 100 µL of ROS Assay Stain solution was added to each 1 mL of cell culture in flow cytometry tubes and well mixed. Then, the cells were incubated in a 5% CO_2_ atmosphere at 37 °C for 60 min. After this process, the cells were analyzed by flow cytometry, using a 488 nm excitation and green emission for ROS (BL1 channel).

### 2.8. Evaluation of Caspase 3/7 Activity

In total, 300 µL of both cell cultures were transferred in flow cytometry tubes; then, 20 µL of a Magic Red^®^ Caspase-3/7 Substrate—MR-(DEVD)_2_—solution was well-mixed with the cells. Next, 20 µL of PI was added. After incubation, 1 mL FCB was added. Then, the early stages of cell apoptosis by activating caspases 3/7 (DEVD-ases) [55,56] were analyzed through flow cytometry using a 488 nm excitation, red emission for MR-(DEVD)_2_—BL3 channel, and orange emission for PI—BL2 channel.

### 2.9. Cell Cycle Analysis

A cell culture volume of 1 mL was washed in FCB, introduced into flow cytometry tubes, and fixed with 50 µL ethanol for 30 min [51]. Next, the cells were treated with PI (20 µg/mL) and RNase A (30 µg/mL) and incubated at room temperature, in darkness, for 30 min [51]. Then, 1 mL FCB was added, and the cell cycle distribution was detected at the flow cytometer in the following conditions: a 488 nm excitation and orange emission for PI (BL2 channel) [51].

### 2.10. Evaluation of Nuclear Condensation and Lysosomal Activity

Magic Red^®^ Caspase-3/7 Assay Kit contains Hoechst 33342 stain (200 μg/mL) and acridine orange (AO, 1.0 µM). Hoechst 33342 is a cell-permeant nuclear stain [57]; when it is linked to double chain DNA, it emits blue fluorescence, highlighting condensed nuclei in apoptotic cells. Acridine orange is a chelating dye that can be used to reveal lysosomal activity [58,59]. Therefore, 300 µL of each cell culture was introduced in flow cytometry tubes; then, 2 µL of Hoechst 33342 stain was added, and the cells were mixed well [51]. After these operations, 50 µL of AO (1.0 µM) was added; the cells were incubated at room temperature in darkness for 30 min. After incubation, 1 mL FCB was added; the cells were examined at the flow cytometer under the following conditions: an excitation of 488 nm, the UV excitation, and blue emission for Hoechst 33342 (VL2), and green emission acridine orange (BL1 channel) [51].

### 2.11. Annexin V-FITC Apoptosis Assay

The normal blood cells and CLS-354 tumor cells were incubated in flow cytometry tubes with 2 µL Annexin V-FITC and 2 µL PI (20 µg/mL) for 30 min, at room temperature, in darkness. After incubation, 1 mL of FCB was added. All viable cells, early apoptotic cells, late apoptotic cells, and necrotic cells were examined at a flow cytometer using the following conditions: an excitation of 488 nm and the following two emission types: green—for Annexin V-FITC (BL1 channel) and orange—for PI (BL2 channel) [51].

### 2.12. Evaluation of Cell Proliferation

Volumes of 1 mL of both cell cultures were incubated with 50 µM EdU (500 µL) at 37 °C for 2 h. Then, both cell types were fixed with 4% paraformaldehyde in PBS (100 µL) and permeabilized with Triton X-100 (100 µL). After washing in 3% buffer sodium azide (BSA) and centrifuging at 300 rpm for 5 min at 4 °C, the cells were incubated with a reaction mix (500 µL) for 30 min at room temperature in darkness. Then, they were washed in permeabilization buffer and centrifuged at 300 rpm for 5 min at 4 °C. After these procedures, 1 mL FCB was added, and the cells were examined by flow cytometry, using a 488 nm excitation and green emission for EdU-iFluor 488 (BL1).

### 2.13. Antimicrobial Activity Evaluation by Resazurin-Based 96-Well Plate Microdilution Method

#### 2.13.1. Inoculum Preparation

The direct colony suspension method (CLSI) was used for preparing the bacterial inoculum. Thus, bacterial colonies selected from a 24 h agar plate were suspended in an MHA medium. The bacterial inoculum was accorded to the 0.5 McFarland standard, measured at Densimat Densitometer (Biomerieux, Marcy-l’Étoile, France) with around 10^8^ CFU/mL (CFU = colony-forming unit). The fungal inoculum was prepared using the same method, adjusting the RPMI 1640 with fungal colonies to the 1.0 McFarland standard, with 10^6^ CFU/mL.

#### 2.13.2. Samples and Standards

F-UBE-HPMC was dissolved in 1 mL of diluted phosphate buffer. As standards, Ceftriaxone (Cefort 1g Antibiotice SA, Iasi, Romania) solutions 30 mg/mL and 122 mg/mL in distilled water were used for bacteria. The Cefort powder was weighted at Partner Analytical balance (Fink & Partner GmbH, Goch, Germany) and dissolved in distilled water. Terbinafine solution 10.1 mg/mL (Rompharm Company S.R.L., Otopeni, România) was selected as standard for *Candida* sp2.13.3. Resazurin-Based 96-Well Plate Microdilution Method

All successive steps were performed in an Aslair Vertical 700, laminar flow, microbiological protection cabinet (Asal Srl, Cernusco, MI, Italy). In four 96-well plates, we performed seven serial dilutions, adapting the protocol described by Fathi et al. [60] and Elshikh et al. [61].

The 96-well plates were incubated for 24 h at 37 °C for bacteria and 35 °C for yeasts in a My Temp mini Z763322 Digital Incubator (Benchmark Scientific Inc., Sayreville, NJ, USA).

#### 2.13.3. Reading and Interpreting

After 24 h incubation, the colors that appeared in 96-well plates were examined to see the differences between the standard and samples [62]. Moreover, they were examined at the Smart LED Illuminator (Kaneka Eurogentec S.A., Seraing, Belgium) at 470 nm in blue light. The sample’s active concentrations were compared with the standard antibiotic ones. For yeasts, the color chart of the Resazurin dye reduction method was used [63,64]. 

### 2.14. Data Analysis

All analyses were performed in triplicate, and the obtained results were presented as means values ± standard deviation (SD). Our results are presented as percent (%) of cell and nuclear apoptosis, caspase 3/7 activity, autophagy, cell cycle, DNA synthesis, and count (×10^4^) of oxidative cellular stress after flow cytometry analyses were performed with SPSS v. 23 software, 2015 (IBM, Armonk, NY, USA). The Levene test was analyzed for homogeneity of variances of samples. Paired t-test was used to establish the differences between samples and controls, and *p* < 0.05 was considered statistically significant. The principal component analysis (PCA) was performed using XLSTAT 2022.2.1. by Addinsoft (New York, NY, USA) [65].

## 3. Results

### 3.1. Organoleptic Characteristics of Mucoadhesive Oral Patches

The composition and properties of the developed F-UBE-HPMC and R mucoadhesive oral patches are displayed in Table 1.

The manufacturing process led to defined concentrations of UBE in mucoadhesive oral patches formulation, each F-UBE-HPMC enclosing 312 µg UBE, with total phenols content of 178.849 µg and 33.924 µg usnic acid.

Both formulations lead to a homogenous, thin, easy-to-peel, with a uniform, smooth, and glossy surface patch (Appendix A).

The Reference (R, Appendix A) are colorless; F-UBE-HPMC has a green-faint brown color (Appendix A); both R and F-UBE-HPMC are transparent (Appendix A).

The patches’ organoleptic characteristics are highly dependent on the active ingredient state. F-UBE-HPMC maintains the characteristic color of UBE and withstands normal handling and cutting processes without air bubbles, cracks, or imperfections.

### 3.2. Physico-Chemical Analysis of the Mucoadhesive Oral Patches

#### 3.2.1. SEM Morphology

Scanning electron microscopy (SEM) was performed to achieve the patch morphology. Figure 1 shows their surface morphology. The R surface is denser, containing a few small elongated-shaped protrusions (Figure 1a), and the F-UBE-HPMC one is relatively smooth (Figure 1b).

#### 3.2.2. Atomic Force Microscopy

The AFM images are displayed in Figure 1c,d.

The Reference (R) is flat and uniform (repetitive surface features), as pointed out by the arbitrary line scan (surface profile line) depicted below the AFM image (Figure 1c) with a vertical height of ~10 nm (see the *Y*-axis of the line scan from −5 to 5 nm). Due to the presence of small agglomerations (in the form of “hills”-see the black arrows in Figure 1e), not exceeding 70 nm in the vertical direction, the peak-to-valley parameter (Rpv) is 86.5 nm (Figure 1c,f). In comparison, the global root mean square (RMS) roughness of the same area is 3.2 nm (Figure 1e). Random superficial nanometric-sized pores, visible as dark blue spots, are also seen in Figure 1c.

The UBE incorporation changes the Reference morphology (Figure 1d). The F-UBE-HPMC surface becomes more compact; the superficial pores are covered at the nanometric scale. Instead, a few larger pits (see the red arrows in Figure 1d) appeared on the surface, with more than 100 nm in diameter. The RMS roughness of the image from Figure 1d,e is 20.2 nm, while the peak-to-valley global parameter is 240.3 nm, three times more than Reference (Figure 1f). The Rpv and RMS roughness along the line scans are illustrated in Figure 1g,h.

#### 3.2.3. FTIR Spectra

The FTIR spectra for both mucoadhesive oral films (Reference and F-UBE-HPMC) are displayed in Figure 2a,b.

In Figure 2a, the R spectrum shows an absorption band at 3460 cm^−1^ assigned to the stretching frequency of the HPMC’s hydroxyl (-OH) group. Another band at 1345 cm^−1^ is due to the bending vibration of -OH. Other stretching vibration bands related to C-H and C-O were observed at 2926 cm^−1^ and 1058 cm^−1^, respectively. The characteristic vibration peaks associated with HPMC appeared at 1455 cm^−1^ related to the methoxy (-OCH_3_) group and at 946 cm^−1^, corresponding to the pyranose ring [66].

On the other hand, the FTIR spectrum of the F-UBE-HPMC (Figure 2b) shows some peaks found in Reference one, with lower intensities. The peaks are shifted to a lower frequency due to UBE interaction with the polymer matrix.

#### 3.2.4. X-ray Diffractograms

The X-ray diffractograms of R and F-UBE-HPMC are presented in Figure 2c. The X-ray diffraction patterns of the Reference and F-UBE-HPMC (Figure 2c) show that both mucoadhesive oral patches exhibited the characteristic diffraction peaks of HPMC at 2θ = 8° and 2θ = 20° [67]. A careful analysis of their XRD patterns indicates that the amorphous region peak (2θ = 8°) and crystalline region (2θ = 20°) decrease in their intensities due to the UBE blending in the polymer matrix in the case of F-UBE-HPMC (Figure 2c).

#### 3.2.5. Thermogravimetric Analysis

Thermogravimetric and differential thermal analysis aimed to characterize the patch’s thermal behavior and stability. Both patches (Reference and F-UBE-HPMC) exhibit a similar behavior upon heating from 25 to 600 °C (Figure 2d). A 0.8–2.5% weight mass loss occurs on heating up to ~100 °C, which can be associated with the loss of residual solvent and physisorbed water. The decomposition process of the organic compounds occurs in two distinct steps, between 200–400 °C and 400–550 °C. Each decomposition step is accompanied by an exothermic thermal effect (Figure 2d). The mass losses associated with the solvent loss and first and second organic decomposition steps are presented in Table 1 (thermal parameters).

Figure 2d and Table 1 (thermal parameters) indicate that the first stage of both R and F-UBE-HPMC starts at a temperature below 100 °C and is due to the loss of solvent and adsorbed water. The second stage begins from 220 °C to 390 °C, and this stage corresponds to ~85% for R and ~84% for F-UBE-HPMC patches’ weight loss. The third stage, with a maximum at *T*_max_ = 456.8 °C for R and *T*_max_ = 466.2 °C for F-UBE-HPMC, was due to the decomposition of the different organic components. The F-UBE-HPMC thermal stability is higher than the Reference one.

### 3.3. Pharmacotechnical Evaluation of Mucoadhesive Oral Patches

The results of the pharmacotechnical evaluation of F-UBE-HPMC and Reference are presented in Table 1 (pharmacotechnical properties). 

The mucoadhesive oral patches’ weight varies depending on the state of the active ingredient and its dispersion method. No significant differences were registered between F-UBE-HPMC and R. Moreover, considering the variability between the patches of the same series, a remarkable uniformity is noticed.

The F-UBE-HPMC thickness (mm) is similar to Reference one: 0.082 ± 0.003 vs. 0.081 ± 0.002, *p* > 0.05. Low SDs in thickness indicate no significant differences within each patch type.

Both formulations exhibited a great folding endurance, with values above 300, proving suitable flexibility. Their flexibility is induced by the plasticizer used in the formulation and the patch-forming polymer [68].

Regarding the mechanical properties of the F-UBE-HPMC and Reference, it can be noticed that the differences between the formulations are not so significant; this could be predictable as they contain identical amounts of HPMC and PEG 400. Also, it is remarked that the UBE-loaded mucoadhesive oral patches display a higher elongation (51.26%) and a lower tensile strength (2.55 kg/mm^2^) than their References (49.47%, respectively 2.83 kg/mm^2^), proving the influence of the active ingredients on their resistance and elasticity. F-UBE-HPMC elongation and tensile strength are adequate to resist during handling [69].

The F-UBE-HPMC moisture content was 6.24%, with minor differences compared to Reference (6.02%). Both patches present suitable humidity. The moisture is due to the solvent system used in the formulation or to the ingredients’ hygroscopic properties, especially the plasticizer [70]. A moderate amount of moisture is needed in mucoadhesive oral patches to ensure their elasticity and protection from being brittle, dry, and easy to break [71].

F-UBE-HPMC has a neutral pH value close to the oral cavity, ensuring good tolerability with no possible irritation of the buccal mucosa. It can be noticed that the active ingredients did not modify the pH of the matrix system, as the tested UBE-loaded formulation displayed a similar pH value to the Reference one.

F-UBE-HPMC reveals an in vitro disintegration time in a simulated saliva medium of 130 s. There are no differences between F-UBE-HPMC and R, suggesting that the UBE load does not influence the patches’ disintegration performance. However, these results offer only orientation data considering that when applied to the oral mucosa, the patches are immobilized, and the fluid medium is secreted in low quantity.

The swelling rate over the 6 h of the study is presented in Figure 2e.

No significant differences between the UBE-loaded patch and Reference are revealed regarding the swelling ratio. The F-UBE-HPMC (272%) displays a lower swelling behavior than R (286%); the UBE state and dispersion determine it. We can observe that the swelling index increases more in the first 4 h, then the growth is slower, the differences between 330 and 360 min being insignificant. It is noticeable that the swelling rate increases linearly in the first 4 h, with around 20% every 30 min. The equilibrium state of swelling was reached at 240 min, and then the swelling was minor. The use of UBE does not considerably influence the swelling ability. No patch eroded after 6 h, and no swelling could be detected after this time.

Regarding the mucoadhesion time, F-UBE-HPMC has a retention time of 102 min on the oral mucosa, the Reference one being 106 min. Generally, the active ingredient dispersion in the polymer base significantly influences the bioadhesive behavior. The retention time depends on the film-forming polymer’s and the plasticizer’s retention properties; it is also highly controlled by the ratio between them [72].

### 3.4. BSL Assay

After 24 h, all the larvae were alive, swimming, and showing normally visible movements. After 48 h, 35.89% of larvae were active, and 12.82% were in the sublethal stage; the registered mortality was 51.28%. We investigated them under a microscope to observe the changes after 24 and 48 h of exposure. All these microscopic images are presented in Figure 3a–p.

The brine shrimp larvae could not feed; after exposure for 24 h to F-UBE-HPMC they have a lower amount of food in the digestive tract (Figure 3e–h) than the negative control (Figure 3a–d). Because of this, they have not even passed to a higher larval stage. After 48 h, the high mortality rate recorded is due to starvation. Compared to blank (Figure 3i–l), the exposed larvae have an empty digestive tract (Figure 3m–p). The lack of nutrients leads to cell and tissue destruction, finalized with the death of *Artemia* nauplii.

In addition, at the intracellular level, FM images (Figure 3q–x) show activated lysosomes in cell death processes (Figure 3w,x).

### 3.5. In Vitro Analysis of the Biological Effects of Mucoadhesive Patches on Human Blood Cell Cultures and Oral Cancer Cell Line CLS-354

Our work continues the preliminary studies of *U. barbata* (L) dry extracts on normal and tumor cells [51,54]. In the present study, *U. barbata* dry extract in 96% ethanol was loaded into mucoadhesive patches (F-UBE-HPMC). Thus, we aim to explore the F-UBE-HPMC biological mechanisms implied in oxidative stress, caspase 3/7 activity, cell cycle, nuclear shrinkage, autophagy, apoptosis, and DNA synthesis in blood cells and CLS-354 tumor cells.

#### 3.5.1. Evaluation of Total ROS Activity

F-UBE-HPMC induced oxidative cellular stress (expressed as total ROS) in blood cell cultures and CLS-354 tumor cell lines (Figure 4).

Significant elevations of ROS levels were reported in the blood cells treated with F-UBE-HPMC compared to both controls as follows: 863.33 × 10^4^ ± 32.14; vs. C1: 242.00 × 10^4^ ± 2.00; *p* < 0.01; C2UA: 846.66 × 10^4^ ± 5.77, *p* ≥ 0.05 (Figure 4a–c,g,i).

The F-UBE-HPMC induced an intense ROS production in OSCC cells, compared to 1% DMSO and 125 µg/mL UA as follows: 1516.66 × 10^4^ ± 105.98; vs. 15.66 × 10^4^ ± 4.04; 966.66 × 10^4^ ± 57.73, *p* < 0.01 (Figure 4d–f,h,j).

#### 3.5.2. Evaluation of Caspases 3/7 Activity

Caspase 3 controls DNA fragmentation and morphological changes in apoptosis. In contrast, caspase 7 appears to be more important to the loss of cellular viability. Thus, the combined role of both caspases is essential in mitochondrial apoptotic events [73]. To examine the pro-apoptotic effects of F-UBE-HPMC through the caspase 3/7 signaling pathway in normal blood cells and CLS-354 tumor cells, we have determined the enzymatic activity of caspase 3/7 by flow cytometry (Figure 5).

Caspase 3/7 activity in blood cell cultures after 24 h of treatment with F-UBE-HPMC reported significantly lower values compared to both controls: 16.16 ± 2.40 vs. C1: 29.26 ± 1.97; C2UA: 44.74 ± 0.41, *p* < 0.01 (Figure 5a–c,g).

In CLS-354 tumor cells, the biochemical cascade of reactions implied in the proapoptotic signal induced by F-UBE-HPMC is slowly higher than the negative control and lower than positive control: 24.84 ± 3.65; vs. 21.88 ± 5.09; 27.02 ± 1.64, *p* ≥ 0.05 (Figure 5d–f,h).

#### 3.5.3. Cell Cycle Analysis

Using propidium iodide/RNase stain for DNA content allowed us to explore the effects of F-UBE-HPMC on cell cycle distribution in normal blood cells and CLS-354 tumor cells (Figure 6).

The F-UBE-HPMC induced a cell cycle arrest in the G1/G0 phase (90.60 ± 0.79) compared to negative control (88.52 ± 0.54, *p* < 0.05) and decreased DNA synthesis (1.91 ± 0.65 vs. 4.76 ± 0.68, *p* < 0.01) in normal blood cells (Figure 6a–c,g,i).

In CLS-354 tumor cells, F-UBE-HPMC determined a higher cell cycle arrest in the G0/G1 phase than the positive control: 93.03 ± 3.13 vs. 90.05 ± 3.45, *p* < 0.01 (Figure 6d–f,h,j). Significantly lower values of DNA synthesis were registered in CLS-354 tumor cells after treatment with F-UBE-HPMC compared to 1% DMSO: 3.58 ± 0.80 vs. 5.47 ± 0.83, *p* < 0.01 (Figure 6d,e,h,j).

#### 3.5.4. Nuclear Condensation and Lysosomal Activity

Apoptotic cells display highly condensed pyknotic nuclei, stained with Hoechst 33342. Acridine orange highlighted the lysosomal activity after F-UBE-HPMC treatment on normal blood cells and CLS-354 tumor cells, as shown in Figure 7.

After 24 h of treatment, the influence of the F-UBE-HPMC on nuclear shrinkage in normal blood cells was substantially lower (7.70 ± 0.80) reported to 1% DMSO: 24.50 ± 2.21, *p* < 0.01 (Figure 7a,b,m). However, they are significantly higher than the 125 µg/mL UA: 3.19 ± 0.30, *p* < 0.05 (Figure 7a,c,m).

The F-UBE-HPMC-induced autophagy had considerably diminished values in normal blood cells, compared to controls: 8.32 ± 0.61; vs. C1: 51.30 ± 3.25; C2UA: 27.05 ± 1.52, *p* < 0.01 (Figure 7g–i,m).

By flow cytometry examination, Hoechst 33342/acridine orange dual stained cells revealed chromatin condensation (NS) and autophagy (A) in OSCC cell lines exposed to F-UBE-HPMC for 24 h. Both processes were significantly intensified compared to 1% DMSO. The NS values were 25.29 ± 1.35 vs. 16.11 ± 3.11, *p* < 0.05 (Figure 7d,e,n) and A levels of 29.75 ± 1.12 vs. 12.57 ± 0.92, *p* < 0.01 (Figure 7j,k,n).

However, F-UBE-HPMC had significantly lower effects on nuclear shrinkage and lysosomal activity than 125 µg/mL UA: NS: 25.29 ± 1.35 vs. 44.03 ± 0.36, *p* < 0.01; A: 29.75 ± 1.12 vs. 53.35 ± 2.63, *p* < 0.01 (Figure 7d,f,j,l,n).

#### 3.5.5. Annexin V-FITC Apoptosis Assay

Cell apoptosis by externalizing phosphatidyl serine—as evidenced by annexin V-FITC/PI stain—triggered by F-UBE-HPMC was determined by flow cytometry based on morphology and cell membrane integrity in normal blood cells and CLS-354 tumor cells (Figure 8).

After 24 h of treatment, in normal blood cells, F-UBE-HPMC did not induce early cell apoptosis (EA); the cell viability (V) had significant differences compared to C2UA. The following values were registered: EA—0.00 ± 0.00 vs. 37.04 ± 0.66, *p* < 0.01; V—98.76 ± 1.13 vs. 61.43 ± 0.88, *p* < 0.01 (Figure 8a,c,g).

Similar results were obtained at F-UBE-HPMC compared to 125 µg/mL UA regarding EA of tumor cells: 0.00 ± 0.00 vs. 12.92 ± 1.35, *p* < 0.01. Only UA induced EA in OSCC cells. Generally, the viability of tumor cells exposed to F-UBE-HPMC had considerably higher values than positive control: 99.36 ± 0.61 vs. 54.05 ± 1.35, *p* < 0.01 (Figure 8d,f,h).

#### 3.5.6. Cell Proliferation

EdU (5-ethynyl-2’-deoxyuridine) directly measured DNA synthesis in normal blood cells and CLS-354 tumor cells treated with F-UBE-HPMC. Simultaneously, this test permitted the evaluation of DNA fragmentation as a sub G0/G1 phase corresponding to apoptotic cell fraction [74]. The obtained results are presented in Figure 9.

In normal blood cells, F-UBE-HPMC considerably diminished DNA synthesis compared with both controls as follows: 2.75 ± 0.21 vs. 10.36 ± 1.21 (C1) and 6.49 ± 1.25 (C2UA), *p* < 0.05. DNA fragmentation implied in apoptosis significantly increased compared to 125 µg/mL UA: 2.49 ± 0.50 vs. 0.00 ± 0.00, *p* < 0.05 (Figure 9a–c,g,i).

On the other hand, in CLS-354 tumor cells, F-UBE-HPMC blocked DNA synthesis compared to both controls as follows: 0.00 ± 0.00 vs. 12.44 ± 2.80 (C1) and 3.14 ± 0.50 (C2UA), *p* < 0.05. However, the apoptotic cell fraction represented by fractional DNA (sub G0/G1 phase) [74] had lower values reported to 1% DMSO: 1.54 ± 0.70 vs. 15.18 ± 2.17, *p* < 0.05 (Figure 9d–f,h,j).

#### 3.5.7. Principal Component Analysis

The principal component analysis (PCA) [75] was realized for F-UBE-HPMC and both controls (C1-DMSO and C2UA) and variable parameters measured in normal blood cells and CLS-354 OSCC tumor cells, according to the correlation matrix and PCA-correlation circle from Appendix A.

The PCA results are illustrated in Figure 10.

The two principal components explained the total data variance, with 64.26% attributed to the first (PC1) and 35.74% to the second (PC2). The PC1 was associated with controls (C1-DMSO and C2UA), caspase 3/7 activity in CLS-354 tumor cells, and ROS levels in normal blood cells. PC2 was related to F-UBE-HPMC mucoadhesive oral patches, ROS levels in CLS-354 tumor cells, and caspase 3/7 activity in normal blood cells.

In normal blood cells, caspase 3/7 activity shows a high positive correlation with autophagy (*r* = 0.998, *p* < 0.05), DNA synthesis (*r* = 0.999, *p* < 0.05) and necrosis (*r* = 0.970, *p* > 0.05) and a moderate one with nuclear condensation (*r* = 0.758, *p* > 0.05). This pathway is highly negatively correlated with oxidative stress and cell cycle arrest in G0/G1 (*r* = −0.885, respectively *r* = −0.970, *p* > 0.05). Figure 10 also indicates that ROS level highly positively correlates with cell cycle arrest in G0/G1 *(r* = 0.972, *p* > 0.05); it reports a low correlation with (*r* = 0.479, *p* > 0.05). Oxidative stress shows a high negative correlation with DNA synthesis (*r* = −0.882, *p* > 0.05), necrosis (*r* = −0.971, *p* > 0.05), nuclear condensation (*r* = −0.975, *p* > 0.05), and autophagy (*r* = −0.911, *p* > 0.05).

In CLS-354 tumor cells, the effector caspase 3/7 activity is substantially positively correlated with autophagy *(r* = 0.984, *p* > 0.05), nuclear condensation *(r* = 0.960, *p* > 0.05), early apoptosis *(r* = 0.819, *p* > 0.05), late apoptosis *(r* = 0.819, *p* > 0.05) and necrosis *(r* = 0.826, *p* > 0.05) and moderately correlated with ROS levels *(r* = 0.692, *p* > 0.05). This previously mentioned mechanism negatively correlates with DNA fragmentation in the sub G0/G1 phase (high, *r* = −0.896, *p* > 0.05), DNA synthesis (moderate, *r* = −0.776, *p* > 0.05), and cell cycle arrest in G0/G1 (low, *r* = −0.371, *p* > 0.05). The oxidative stress is considerably negatively correlated with DNA synthesis (*r* = −0.992, *p* > 0.05) and sub G0/G1 phase (*r* = −0.940, *p* > 0.05), and low positively correlated with early and late apoptosis, necrosis, nuclear condensation, and autophagy (*r* = 0.152–0.553, *p* > 0.05).

A general data analysis for the good visualization of F-UBE-HPMC and controls activity on normal and tumor cells is illustrated in Appendix A. Appendix A shows that F-UBE-HPMC and the positive control (UA of 125 µg/mL) exhibit a higher activity on tumor cells than on normal blood cells (Appendix A). The UBE-loaded mucoadhesive oral patches and UA diminish the blood cell damage induced by 1% DMSO through caspase 3/7 activity, nuclear condensation, and autophagy which trigger necrotic processes (Appendix A). Moreover, F-UBE-HPMC has a higher protective action on normal cells than UA. On CLS-354 tumor cells, the UBE-loaded mucoadhesive patches induced the most elevated oxidative stress and complete inhibition of DNA synthesis (Appendix A). Usnic acid, the main phenolic secondary metabolite of *Usnea* sp., exhibited the highest antitumor activity. These data correlation and interpretation established the places of F-UBE-HPMC and both controls (C1-DMSO and C2UA) in the PCA-biplot (Figure 10), highlighting the corresponding processes triggered in CLS-354 cancer cells and normal blood cells.

### 3.6. Antimicrobial Activity

Data registered in Table 2 display the used in microdilutions of standard antibiotic (CTR) and antifungal (TRF) drugs and F-UBE-HPMC. Data from Table 2 show that the colors of standard antibiotics correlate with their inhibiting power and are directly proportional to their concentration. We can observe CTR dose-dependent inhibitory activity on both tested bacteria. However, *S. aureus* sensibility at CTR is higher than *P. aeruginosa.*

Contrariwise, the inhibitory activity of F-UBE-HPMC is higher on *P. aeruginosa* than on *S. aureus.* Therefore, the inhibitory activity of F-UBE-HPMC of [5.2–0.65] mg/mL against *S. aureus* is similar to CTR of [0.75–0.093] mg/mL. On *P. aeruginosa,* F-UBE-HPMC of [5.2–0.325] mg/mL acts similarly with CTR of [1.998–1.599] mg/mL; lower concentrations of [0.162–0.081] mg/mL have a similar effect with CTR of [0.093–0.046] mg/mL.

Data from Table 2 shows that TRF had a fungicidal effect on both *Candida* sp. [64] and in the entire microdilutions domain. 

On *C. albicans*, the F-UBE-HPMC of [5.2–0.65] mg/mL acts similarly to TRF. The lower dilutions of [0.325–0.081] mg/mL exhibit a significantly diminished effect, inducing a moderate to fast proliferation of *C. albicans* colonies (Table 2). 

After 24 h of incubation with the first two dilutions of F-UBE-HPMC [5.2–2.6] mg/mL, *C. parapsillosis* fungal cells were partially dead [64]. The following F-UBE-HPMC concentrations of [1.3–0.081] mg/mL progressively induced low to moderate cell proliferation.

## 4. Discussion

Until the pharmaceutical formulation, our team studied *U. barbata* from the Calimani mountains for almost 6 years, according to current medicinal plant legislation [76,77,78]. In the present work, mucoadhesive oral patches containing *U. barbata* dry ethanol extract were manufactured and analyzed, compared to References (the same formulation without UBE). HPMC in a 15% aqueous dispersion ensured suitable patch toughness. PEG 400—in a 5% concentration of the patch mass—provided an elegant, glossy, smooth appearance and high flexibility. The F-UBE-HPMC homogeneity proved that the active ingredient was adequately incorporated into the polymer matrix due to the miscibility of the vehicles.

Weight and thickness low variations guarantee the efficiency of the formulation and applied method and provide a certain content uniformity. F-UBE-HPMCs have suitable weight and thickness for application to the oral mucosa [45,79,80], and our results are similar to other developed studies on HPMC patches [79].

The patches’ flexibility is essential for easy handling and administration. Semalty et al. [81] proved that mixing HPMC with PEG in 30% of the polymer weight leads to low folding endurance. Thus, PEG used in low concentrations represents the optimal plasticizer. In the present study, PEG 400 in a 5% proportion confirms the excellent flexibility of both patches. The plasticizer reduces patch rigidity by minimizing intermolecular forces [82]. High amounts of plasticizer, due to over-hydration, might diminish the patches’ mucoadhesive properties [83].

In F-UBE-HPMC, UBE is dissolved in the base, thus maintaining its natural structure. The polymer molecular chain disruption induces higher chain mobility, increasing flexibility, and decreased rigidity. Maher et al. [84] proved the influence of the polymer type on the patch tensile strength. Also, it was confirmed that the tensile strength increases with the film-forming polymer concentration. The HPMC 15% dispersed in water leads to developing a strong matrix with a suitable network density. The results show the film-forming agent and the plasticizer’s substantial influence on the patches’ mechanical properties. Their strength is affected by the active ingredient’s nature, concentration, and dispersion type.

The amount of moisture influences the patches’ friability; as proven, both ones display good resistance. Thus, the contained humidity could offer suitable mechanical properties. PEG 400 highlights a substantial hygroscopicity due to its hydrophilic hydroxyl groups interacting with water [35], providing numerous sites for interactions and leading to the patches’ moisture retention. However, HPMC has hydrophilic hydroxypropyl substituents but contains hydrophobic methoxyl groups and does not maintain excessive moisture [85].

All the ingredients influence the patches’ pH values. We aimed to properly select them to obtain a surface pH similar to the buccal one, and F-UBE-HPMCs are biocompatible with the oral mucosa, having a neutral pH.

The disintegration time displayed by both patches intensely depends on the polymer matrix. Shen et al. [86] demonstrated that the patch disintegration time rises with increasing HPMC concentration. The patches’ in vitro behavior was typical for rapid disintegration systems, allowing a fast release of the active ingredient.

The swelling properties are essential for patches’ mucoadhesion. They considerably depend on the diffusivity of water into the polymer [87]. Therefore, it was proven that the disturbance of the polymer chains by including active ingredients in the matrix decreases the water content [88].

The values registered for ex vivo retention time help predict the patches’ in vivo mucoadhesive performance. The obtained results proved to be satisfactory for the studied formulations. Adhesion is enhanced with the hydration increase until an optimal point; overhydration determines the polymer/tissue interface damage and diminishes the bioadhesive force.

The neutral pH and rapid disintegration of F-UBE-HPMC permitted in vitro and in vivo investigation. The *U. barbata* dry ethanol extract was previously studied and proved to have significant pharmacological potential. Its secondary phenolic metabolites were quantified through HPLC-DAD; UBE has a total phenolic content of 573.234 ± 42.308 mg PyE/g, of which 108.742 ± 0.703 mg/g usnic acid, 0.605 ± 0.007 mg/g ellagic acid, and 0.870 ± 0.008 mg/g gallic acid [65]. Moreover, using UHPLC/ESI/MS/MS in negative mode, Salgado et al. [89] identified other secondary metabolites with phenolic structures (depsides, depsidones, diphenyl ethers). We investigated the cytotoxic activity of UBE on brine shrimp larvae [40], normal blood cells [51], and OSCC cancer cells CAL-27 [54], and its antimicrobial activity on various bacterial and fungal strains [90,91,92].

To compare the F-UBE-HPMC bioactivities with the UBE ones, we opted for similar studies: BSL assay for cytotoxicity prescreening, blood cell cultures from the same donor (a member of our research team), and another OSCC cell line (CLS-354). 

The brine shrimp lethality assay is a significant and low-cost antitumor prescreening in anticancer drug discovery [49]. The effects of F-UBE-HPMC on *A. salina* nauplii can be extrapolated to the analysis of their activity on tumor cells [93]. After 24 h of exposure, the brine shrimp larvae were alive with normal movements. However, the microscopic examination of *A salina* nauplii showed that they could not feed. Compared to the control, there is a low food amount in the digestive tract of the exposed larvae for 24 h to the F-UBE-HPMC; therefore, they have not even passed to a higher larval stage. After 48 h, a high level of larvae mortality was registered; it could be extrapolated to tumor cells as a blockage of DNA synthesis and cell cycle arrest in G0/G1 [93,94].

After 24 h of F-UBE-HPMC treatment, the viability of tumor cells was over 99%, similar to BSL assay results. However, the flow-cytometry analyses have shown that UBE-loaded mucoadhesive oral patches trigger apoptotic mechanisms in OSCC cells by a massive increase in ROS level and stimulation of effector caspases 3/7 enzymatic activity. Both mechanisms lead to DNA synthesis blockage, cell cycle arrest in G0/G1, nuclear condensation, and autophagy. Through the pro-oxidant activity, f-UBE-HPMC exhibits a higher activity on tumor cells than on normal blood cells. In normal blood cells, the patches highlighted a substantially lower ROS production. Their protective effects on normal cells diminish the blood cell damage induced by 1% DMSO through necrotic processes and reduce caspase 3/7 enzymatic activity, nuclear condensation, and autophagy. All these results could be explained by the dual redox behavior of usnic acid and other phenolic secondary metabolites (pro-oxidant in tumor cells and antioxidant in normal cells), as proved by numerous studies from the scientific literature [54,95,96,97,98,99,100,101,102,103]. Compared to the patches, the UBE effects were faster, with the larvae and cell viability diminished after 24 h [40,51,54]. Moreover, its activity on tumor cells was noticeable. UBE also proved an effective in vitro wound-healing potential, higher on normal cells than tumor ones [54].

In their previous studies, Jardón-Romero et al. [104], Rafey et al. [105], and Thiyahuddin et al. [106] considered *S. aureus, P. aeruginosa, C. albicans,* and *C. parapsillosis* as the most frequent pathogens responsible for opportunistic oral cavity infections in immunocompromised and older people; we previously examined UBE inhibitory activity against these microbial species [90]. Our results showed that oral mucoadhesive patches loaded with *U. barbata* dry ethanol extract have a dose-dependent inhibitory activity against these microorganisms. Usnic acid and other phenolic constituents from UBE underlie these antimicrobial effects through various mechanisms, acting on many sites at the cellular level [107,108,109]. The results of the F-UBE-HPMC pharmacological potential investigation proved that this pharmaceutical formulation preserved all the properties of UBE as an active ingredient. Moreover, the manufacturing process and polymers used for obtaining UBE-loaded mucoadhesive oral patches did not affect the bioactive secondary metabolites of *U. barbata* dry ethanol extract, being optimal for its therapeutic purposes.

It is known that usnic acid, the main secondary metabolite of *Usnea* sp., displays severe hepatotoxicity. It caused fulminant liver failure when it was used as a fat burner in a complex dietary supplement (LipoKinetix, Syntrax, Cape Girardeau, MO, USA) [110], associated with a recommended daily dose of 300–600 mg. *Usnea* sp is also known in Traditional Chinese Medicine as a liver detoxifier. The common TCM dosages were 6–9 g of dried lichen, corresponding to around 60–120 mg of usnic acid per day [111]; thus, 1768 patches cover the minimal used daily dose.

## 5. Conclusions

In this study, the mucoadhesive oral patches loaded with *U. barbata* dry ethanol extract were prepared using HPMC and PEG 400 for their formulation. The F-UBE-HPMCs were compared to References (the same patches without active lichen extract) through complex physico-chemical and pharmacotechnical procedures, proving their suitability for oral topical administration.

Moreover, F-UBE-HPMC pharmacological potential investigation confirmed an in vitro anticancer activity on oral squamous cell carcinoma and dose-dependent inhibitory effects against the most common bacterial and fungal pathogens implicated in immunosuppressed patients’ oral infections.

The results suggest that the mucoadhesive oral patches loaded with *U. barbata* dry ethanol extract could be a promising phytotherapeutic formulation with potential application in oral medicine.

## Figures and Tables

**Figure 1 antioxidants-11-01801-f001:**
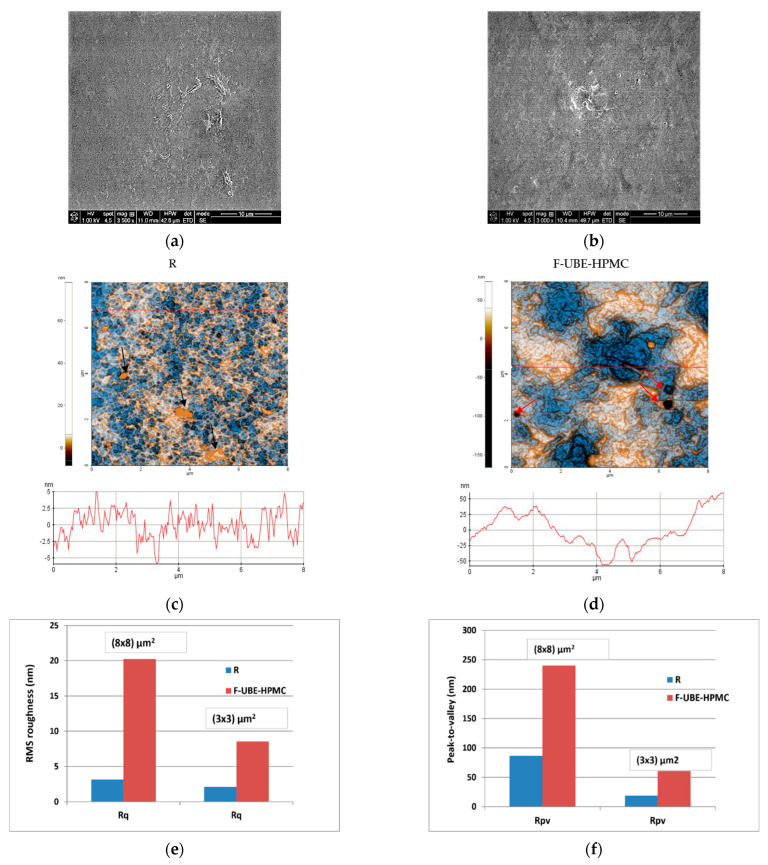
SEM images of R (**a**) and F-UBE-HPMC (**b**); 2D-AFM images (enhanced contrast view) at the scale of (8 × 8) µm^2^ together with representative line scans for R (**c**) and F-UBE-HPMC (**d**). Roughness (Rq) and peak-to-valley (Rpv) parameters for the whole scanned areas at (8 × 8) µm^2^ and (3 × 3) µm^2^ (**e**,**f**) and along the line scans over 8 µm and 3 µm (**g**,**h**). F-UBE-HPMC—mucoadhesive oral patches loaded with UBE; UBE—*U. barbata* dry ethanol extract; R—Reference (mucoadhesive oral patches without UBE).

**Figure 2 antioxidants-11-01801-f002:**
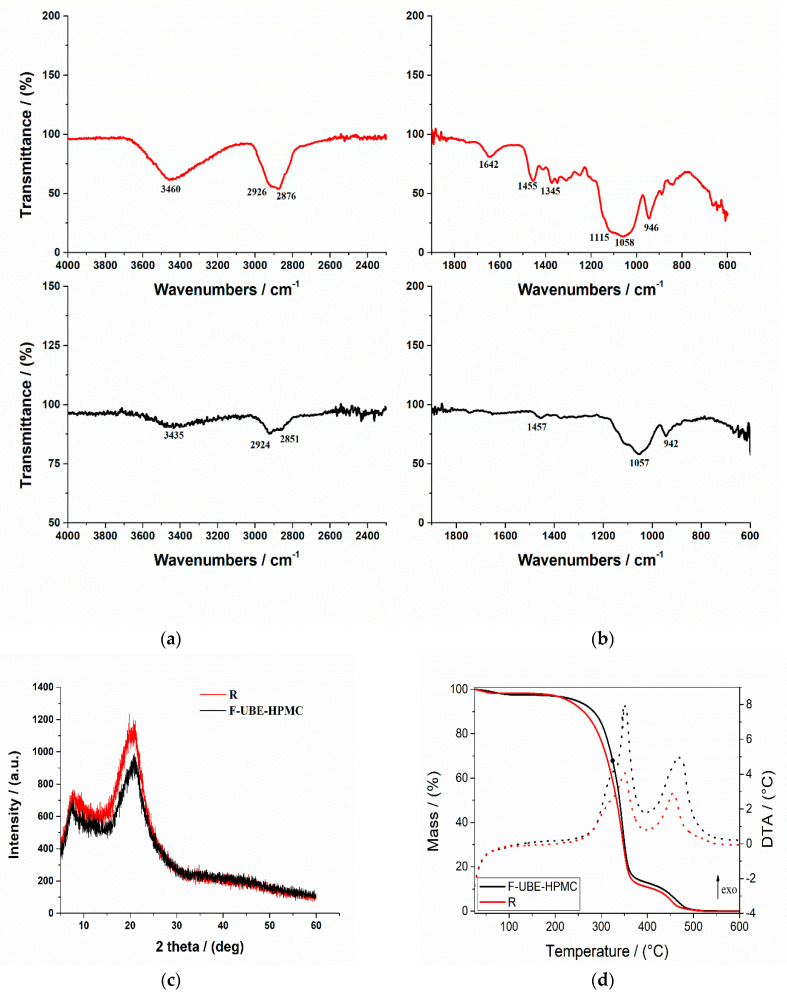
FTIR Spectra of mucoadhesive oral patches (**a**,**b**): Reference (red line) and F-UBE-HPMC (black line) in the range 4000–2200 cm^−1^ (**a**) and 2000–400 cm^−1^ (**b**); X-ray Diffractograms of Reference and F-UBE-HPMC oral mucoadhesive patches (**c**); Thermogravimetric analysis coupled with differential thermal analysis of the mucoadhesive oral patches (**d**). Swelling rate% over 6 h of F-UBE-HPMC versus R (**e**). R—Reference (patch without UBE); F-UBE-HPMC—mucoadhesive oral patches loaded with UBE; UBE—*U. barbata* dry ethanol extract.

**Figure 3 antioxidants-11-01801-f003:**
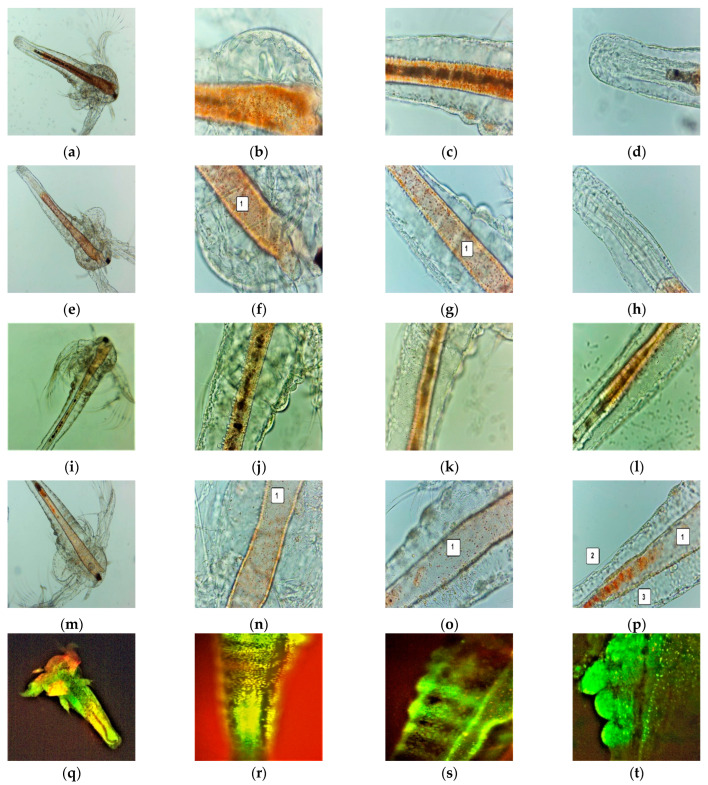
*A. salina* larvae after the exposure to F-UBE-HPMC (**a**–**p**)—microscopic images at 100× (**a**,**e**,**i**,**m**) and 400× (**b**–**d**,**f**–**h**,**j**–**l**,**n**–**p**). After 24 h: blank (**a**–**d**) and F-UBE-HPMC (**e**–**g**); after 48 h: blank (**i**–**l**) and F-UBE-HPMC (**m**–**p**). The following changes can be observed compared to blank: (**f**–**g**,**n**–**p**) a low quantity of food in the digestive tract (1), (**p**) a low detachment of the cuticle from larval tissues (2), the cell damage with large intercellular spaces (3). FM images of *A. salina* larvae after 48 h of exposure at F-UBE-HPMC (**q**–**x**) stained with acridine orange 100× (**q**,**u**) 200× (**r**–**t**,**w**,**x**) and 400× (**f**). (**q**–**t**)—blank; (**u**–**x**)—F-UBE-HPMC. Intracellular lysosomes activated in cell death processes were revealed by the red fluorescence (**w**,**x**). F-UBE-HPMC—mucoadhesive oral patches loaded with *U. barbata* dry ethanol extract.

**Figure 4 antioxidants-11-01801-f004:**
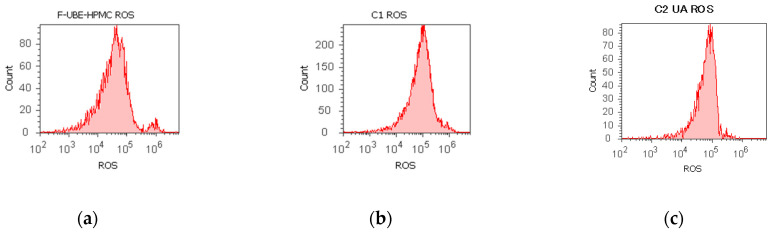
Reactive oxygen species (ROS) levels models in normal blood cells (**a**–**c**) and CLS-354 tumor cells (**d**–**f**) after 24 h treatment with F-UBE-HPMC. UBE-loaded mucoadhesive patches and controls extrapolated on ROS axis (**g**,**h**) in blood cells (**g**) and CLS-354 tumor cells (**h**); F-UBE-HPMC—mucoadhesive oral patch loaded with *U. barbata* dry ethanol extract (**a**,**d**); C1—1% DMSO negative control (**b**,**e**); C2—125 µg/mL UA-positive control (**c**,**f**). Statistical analysis of reactive oxygen species (ROS) in normal blood cells (**i**) and CLS-354 tumor cells (**j**). ** *p* < 0.01 evidence of significant statistical differences between controls and F-UBE-HPMC made by paired samples *t*-test; F-UBE-HPMC—mucoadhesive oral patches loaded with *U. barbata* (L.) dry ethanol extract; C1—negative control with 1% dimethyl sulfoxide (DMSO); C2UA—positive control with 125 µg/mL usnic acid (UA).

**Figure 5 antioxidants-11-01801-f005:**
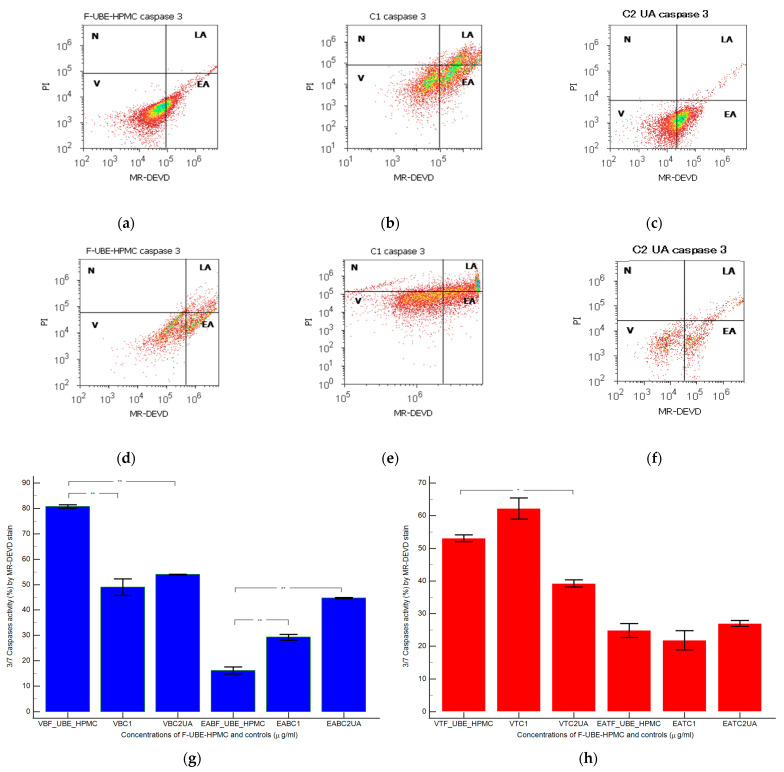
Caspases 3/7 activity models in normal blood cells (**a**–**c**) and CLS-354 tumor cells (**d**–**f**) after 24 h treatment with F-UBE-HPMC. MR-DEVD patterns of F-UBE-HPMC (**a**,**d**), 1% DMSO negative control (**b**,**e**), and 125 µg/mL UA-positive control (**c**,**f**); Statistical analysis of 3/7 caspases activity in normal blood cells (**g**) and CLS-354 tumor cells (**h**). * *p* < 0.05 and ** *p* < 0.01 evidence significant statistical differences between controls and F-UBE-HPMC made by paired samples *t*-test; V—viability; EA—early apoptosis; F-UBE-HPMC—mucoadhesive oral patches loaded with *U. barbata* (L.) dry ethanol extract; C1—negative control with 1% dimethyl sulfoxide (DMSO); C2UA—positive control with 125 µg/mL usnic acid (UA).

**Figure 6 antioxidants-11-01801-f006:**
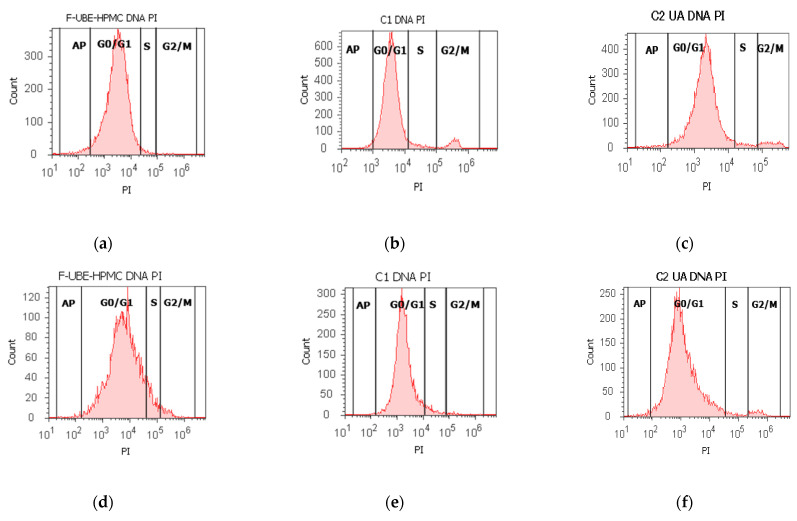
Cell cycle analysis in normal blood cells (**a**–**c**) and CLS-354 tumor cells (**d**–**f**) after 24 h treatment with F-UBE-HPMC. PI/RNase patterns of F-UBE-HPMC (**a**,**d**); 1% DMSO negative control (**b**,**e**); 125 µg/mL UA-positive control (**c**,**f**); F-UBE-HPMC and controls extrapolated on PI axis (**g**,**h**); AP—apoptosis (sub G0/G1); Statistical analysis (**i**,**j**) of G0/G1, synthesis (S), and G2/M phases of the cell cycle in normal blood cells (**i**) and CLS-354 tumor cells (**j**). * *p* < 0.05 and ** *p* < 0.01 represents significant statistical differences between controls and F-UBE-HPMC made by paired samples *t*-test; F-UBE-HPMC—mucoadhesive oral patches loaded with *U. barbata* (L.) dry ethanol extract; C1—negative control with 1% dimethyl sulfoxide (DMSO); C2UA—positive control with 125 µg/mL usnic acid (UA).PI–propidium iodide; S-synthesis of cell cycle phases; F-UBE-HPMC—mucoadhesive oral patches loaded with *U. barbata* (L.) dry ethanol extract.

**Figure 7 antioxidants-11-01801-f007:**
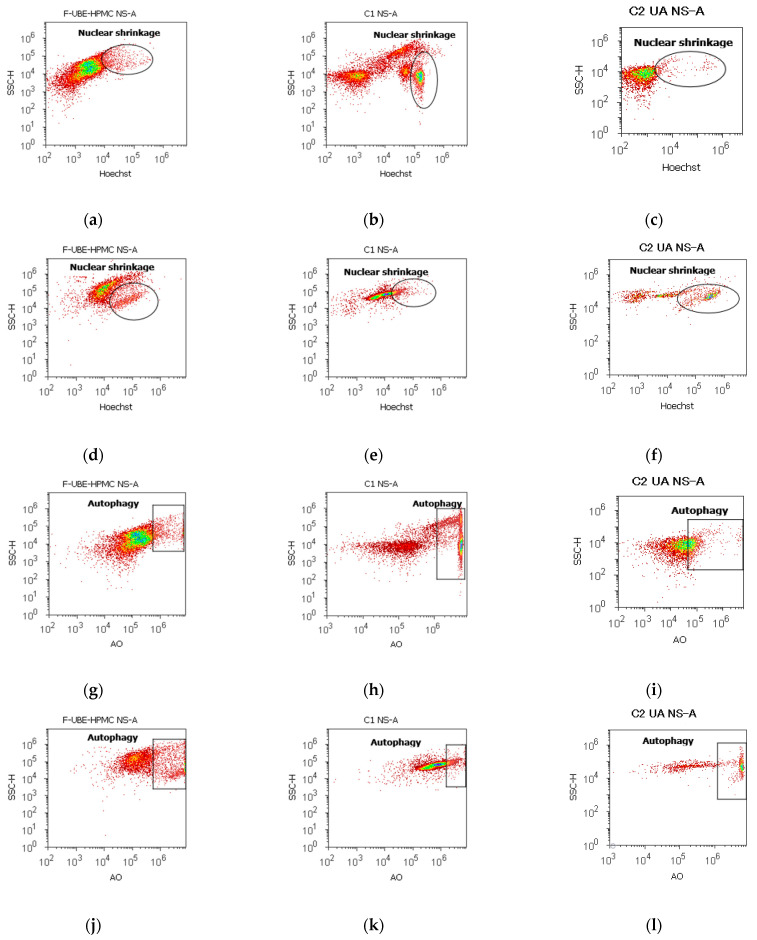
Nuclear shrinkage models (**a**–**f**) in normal blood cells (**a**–**c**) and CLS-354 tumor cells (**d**–**f**), after 24 h treatment with F-UBE-HPMC. Hoechst patterns of F-UBE-HPMC (**a**,**d**); 1% DMSO negative control (**b**,**e**); 125 µg/mL UA-positive control (**c**,**f**); Lysosomal Activity (**g-l**) in normal blood cells (**g**–**i**) and CLS-354 tumor cells (**j**–**l**). Acridine orange patterns of F-UBE-HPMC (**g**,**j**); 1% DMSO negative control (**h**,**k**); 125 µg/mL UA-positive control (**i**,**l**); Statistical analysis of nuclear shrinkage and lysosomal activity (**m,n**) in normal blood cells (**m**) and CLS-354 tumor cells (**n**). * *p* < 0.05 and ** *p* ≤ 0.01 reveal significant statistical differences between controls and F-UBE-HPMC made by paired samples *t*-test; NS—nuclear shrinkage; A—autophagy; F-UBE-HPMC—mucoadhesive oral patches loaded with *U. barbata* (L.) dry ethanol extract; C1—negative control with 1% dimethyl sulfoxide (DMSO); C2UA—positive control with 125 µg/mL usnic acid (UA).

**Figure 8 antioxidants-11-01801-f008:**
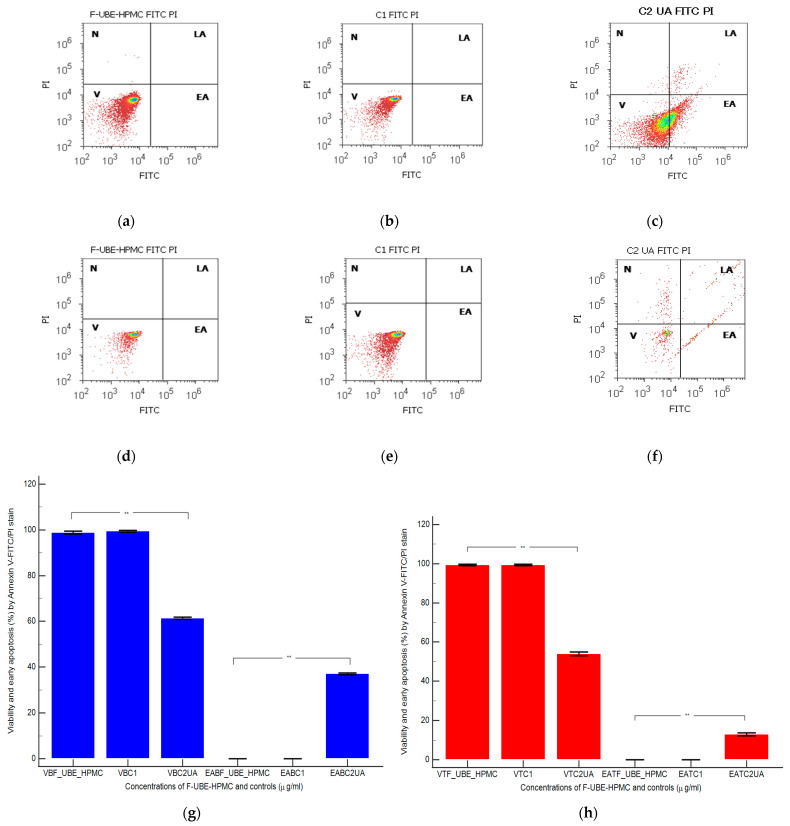
Cell apoptosis models in normal blood cells (**a**–**c**) and CLS-354 tumor cells (**d**–**f**) after 24 h treatment with F-UBE-HPMC. Annexin V-FITC/PI patterns of F-UBE-HPMC (**a**,**d**); 1% DMSO negative control (**b**,**e**); 125 µg/mL UA-positive control (**c**,**f**); F-UBE-HPMC—mucoadhesive oral patches loaded with *U. barbata* (L.) dry ethanol extract. Statistical analysis of cell apoptosis in normal blood cell cultures (**g**) and CLS-354 tumor cell lines (**h**). ** *p* < 0.01 highlights significant statistical differences between controls and F-UBE-HPMC made by paired samples *t*-test; V—viability; EA—early apoptosis; F-UBE-HPMC—mucoadhesive oral patches loaded with *U. barbata* (L.) dry ethanol extract; C1—negative control with 1% dimethyl sulfoxide (DMSO); C2UA—positive control with 125 µg/mL usnic acid (UA).

**Figure 9 antioxidants-11-01801-f009:**
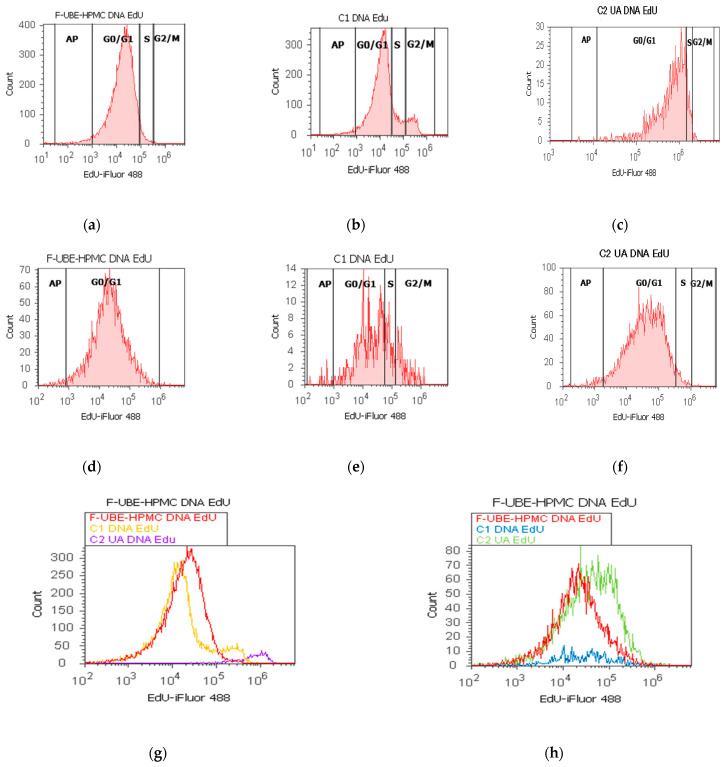
Synthesis (S) and fragmentation of DNA models in normal blood cells (**a**–**c**) and CLS-354 tumor cells (**d**–**f**) after 24 h treatment with F-UBE-HPMC. EdU-iFluor 488 patterns of F-UBE-HPMC (**a**,**d**); 1% DMSO negative control (**b**,**e**); 125µg/mL UA-positive control (**c**,**f**). F-UBE-HPMC and controls extrapolated on EdU-iFluor 488 axis (**g**,**h**); AP—apoptotic cell fraction (sub G0/G1); F-UBE-HPMC—mucoadhesive oral patches loaded with *U. barbata* (L.) dry ethanol extract. Statistical analysis (**i,j**) of DNA synthesis (S) and fragmentation (AP—apoptotic cell fraction, sub G0/G1 phase) in normal blood cell cultures (**i**) and CLS-354 tumor cell lines (**j**). * *p* < 0.05 and *** p* < 0.01 underlines significant statistical differences between both controls and F-UBE-HPMC made by paired samples *t*-test; F-UBE-HPMC—mucoadhesive oral patches loaded with *U. barbata* (L.) dry ethanol extract; C1—negative control with 1% dimethyl sulfoxide (DMSO); C2UA—positive control with 125 µg/mL usnic acid (UA).

**Figure 10 antioxidants-11-01801-f010:**
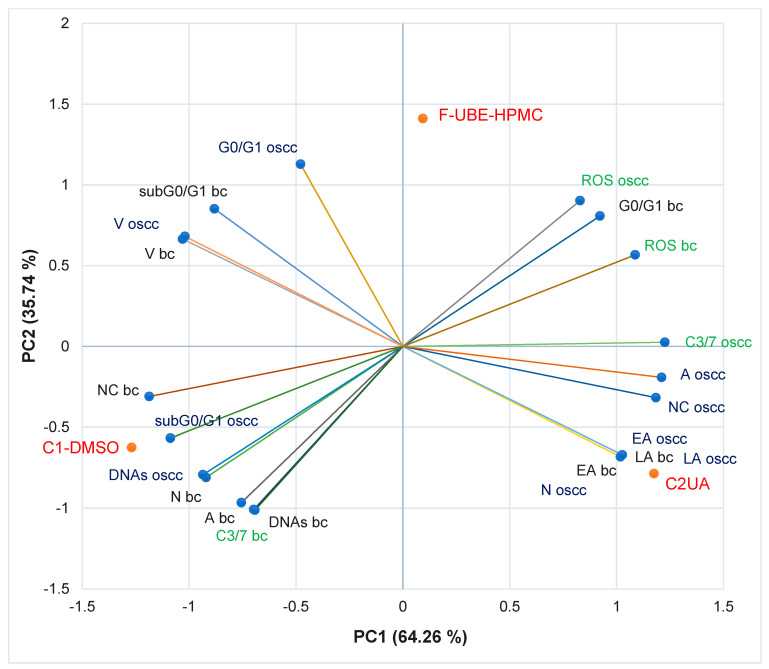
PCA-Correlation biplot between mechanisms (caspase 3/7 activity and cellular oxidative stress) and processes induced by F-UBE-HPMC and both controls (C1-DMSO and C2UA) in normal blood cells (bc) and CLS-354 tumor cells (oscc). F-UBE-HPMC—mucoadhesive oral patches loaded with *U. barbata* (L.) dry ethanol extract; V—viability, EA—early apoptosis, LA—late apoptosis, N—necrosis, NC—nuclear condensation, A—autophagy, DNAs—DNA synthesis, sub G0/G1—apoptotic cell fraction G0/G1—cell cycle arrest in G0/G1 phase, ROS—oxidative stress, C3/7—caspase 3/7 activity.

**Table 1 antioxidants-11-01801-t001:** Composition and properties of mucoadhesive oral patches loaded with *U. barbata* dry extract in 96% ethanol (F-UBE-HPMC) versus References (R).

Variable	F-UBE-HPMC	R
*Ingredients*		
UBE (g)	0.30	-
Ethyl alcohol 96% (*v*/*v*) (g)	10.00	10.00
PEG 400 (g)	5.00	5.00
HPMC 15% water dispersion (*w*/*w*) (g)	84.70	85.00
*Thermal parameters*		
Solvent Mass Loss (%)	2.4	1.8
T (°C)/Mass Loss 1st Decomposition Step (%)	351.7 °C/84.2	466.2 °C/13.4
T (°C)/Mass Loss 2nd Decomposition Step (%)	352.3 °C/85.2	456.8 °C/13.0
*Pharmacotechnical Parameters **		
Weight uniformity (mg)	104 ± 4.31	102 ± 2.55
Thickness (mm)	0.082 ± 0.003	0.081 ± 0.002
Folding endurance value	>300	>300
Tensile strength (kg/mm^2^)	2.55 ± 1.31	2.83 ± 1.25
Elongation %	51.26 ± 1.77	49.47 ± 2.13
Moisture content % (*w*/*w*)	6.24 ± 0.26	6.02 ± 0.14
pH	7.05 ± 0.04	7.03 ± 0.02
Disintegration time (seconds)	130 ± 4.14	131 ± 3.27
Swelling ratio (% after 6 h)	272 ± 6.31	286 ± 4.93
Ex vivo mucoadhesion time (minutes)	102 ± 3.22	106 ± 3.35

UBE—*U. barbata* dry ethanol extract, F-UBE-HPMC—mucoadhesive oral patches with *U. barbata* dry ethanol extract; R—Reference (mucoadhesive oral patches containing the suitable excipients, without UBE); PEG—polyethylene glycol; HPMC—hydroxypropyl methylcellulose; T—temperature; * Expressed as mean value ± SD.

**Table 2 antioxidants-11-01801-t002:** Antibacterial and antifungal activity of UBE-loaded mucoadhesive oral patches.

*Microdilutions*
**Microdilution**	**CTR**	**TRF**	**F-UBE-HPMC**
**30 mg/mL**	**122 mg/mL**	**10.1 mg/mL**	**104 mg/mL**
1	1.5	6.1	0.5	5.2
2	0.75	4.88	0.25	2.6
3	0.375	3.904	0.125	1.3
4	0.187	3.123	0.062	0.65
5	0.093	2.498	0.031	0.325
6	0.046	1.998	0.015	0.162
7	0.023	1.599	0.007	0.081
*Antibacterial activity*
**Dil.**	** *S. aureus* **	** *P. aeruginosa* **
**CTR**	**F-UBE-HPMC**	**CTR**	**F-UBE-HPMC**
**A**	**B**	**A**	**B**	**A**	**B**	**A**	**B**
1		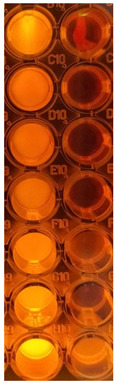	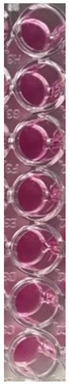	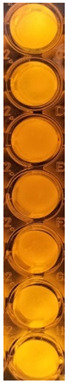		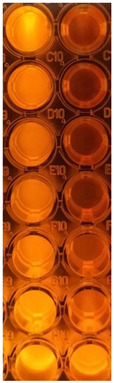	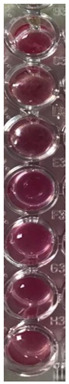	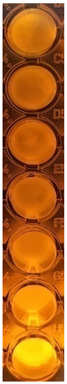
2
3
4
5
6
7
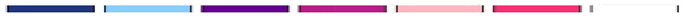 *
*Antifungal activity*
**Dil.**	** *C. albicans* **	** *C. parapsilosis* **	**Color** ******	**Score** ******	**Signification ****
**TRF**	**F-UBE-HPMC**	**TRF**	**F-UBE-HPMC**
**A**	**B**	**A**	**B**	**A**	**B**	**A**	**B**
1		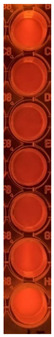	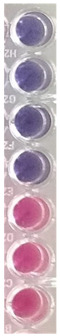	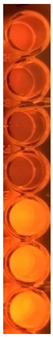		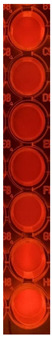	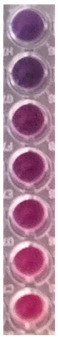	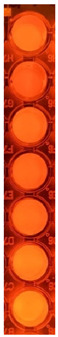	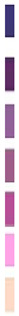	0	Blue—cells are death
2	1	Violet-blue—cells are partially dead
3	2	Violet—cells are alive; no proliferation
4	3	Light-violet—low proliferation
5	4	Dark pink—mode-rate proliferation
6	5	Pink—fast proliferation
7	6	Light pink—very fast proliferation

CTR—ceftriaxone; TRF—Terbinafine; F-UBE-HPMC—mucoadhesive oral patch loaded with UBE; UBE—*U. barbata* dry ethanol extract; * Resazurin dye chart adapted from Madushan et al. [62] as follows: blue—“excellent”; light blue—”very good”; violet—“good”; purple-pink—”moderate”; light pink—“low”; pink—”very low”; white—“no effect”; A. well plates examined by using Resazurin dye chart; B. well plates examined at a wavelength of 470 nm; ** Results interpreting adapted from Bitacura et al. [64]. TRF—Terbinafine, F-UBE-HPMC—mucoadhesive oral patches loaded with UBE (*U. barbata* dry ethanol extract), A—well plates examined through Resazurin color; B—well plates read at a wavelength of 470 nm.

## Data Availability

Data are available in this manuscript.

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
