# Peer review of "Design, Characterization, and Anticancer and Antimicrobial Activities of Mucoadhesive Oral Patches Loaded with *Usnea barbata* (L.) F. H. Wigg Ethanol Extract F-UBE-HPMC"

_antioxidants, 2022, doi:10.3390/antiox11091801_

Round 1
Reviewer 1 Report
- Abstract is too long, very general.
- No detailed profiling for the Usnea barbata (L.) F. H. Wigg Ethanol Extract even by LC/MS and GC/MS
- English, typos, missing spaces such as “225mm mean length, 38mm mean width, ~48N/m, 1.0mg/mL”
- The author should use recent primarily literature rather than too many websites especially the ones that refers to certain products such as “ https://www.plantextrakt.ro/produse/fitoterapice/mira-apa-de-gura-concentrata” and https://pro-natura.ro/produs/perident-plast-10/ and https://www.abcam.com/caspase-37-assay-kit-magic-red-ab270771.html
- Some cited pages are not found https://www.plantextrakt.ro/produse/fitoterapice/dentosept-sup-sup-maxi-spray
- Details on the following clinical trial is needed “The blood samples were collected from a non-smoker healthy donor (B Rh+ blood type), according to Ovidius University of Constanta Ethical approval code 7080/10.06.2021 and Donor Consent code 39/30.06.2021.”
- No consistency of style “In vitro & In vitro)
- Maximum tolerated dose is needs to be considered as well as more safety data is needed.
- Figure 1 is of poor quality
Author Response
Please, see the attachment.

Reviewer 2 Report
Attach file
The manuscript antioxidants-1843633 is entitled: “ Design, Characterization, and Anticancer and Antimicrobial Activities of Mucoadhesive Oral Patches Loaded with Usnea barbata (L.) F. H. Wigg Ethanol Extract F-UBE-HPMC”.
The authors present the Usnea barbata extracts in production of biofilm (mucoadhesive) for use in assay over bacterial and fungus in the oral cavity.
The abstract is complete and with good data obtained and described in the manuscript.
The results obtained from the Usnea barbata extract in ethanol and used in the formation of mucoadhesive (biofilm) are very promising and with the possibility of use in the control of diseases and prevention of the oral cavity.
The graphics and figures are of good quality and well presented.
The references are current.
Minor suggestions I put in the text.
The manuscript is suitable for publication in Antioxidants Journal.

Author Response
Please, see the attachment.

Round 2
Reviewer 1 Report
NA